# Role of miR-133/Dio3 Axis in the T3-Dependent Modulation of Cardiac mitoK-ATP Expression

**DOI:** 10.3390/ijms23126549

**Published:** 2022-06-11

**Authors:** Paola Canale, Giuseppina Nicolini, Letizia Pitto, Claudia Kusmic, Milena Rizzo, Silvana Balzan, Giorgio Iervasi, Francesca Forini

**Affiliations:** CNR Institute of Clinical Physiology, Via G. Moruzzi1, 56124 Pisa, Italy; paola.canale95@libero.it (P.C.); nicolini@ifc.cnr.it (G.N.); letiziapitto@gmail.com (L.P.); kusmic@ifc.cnr.it (C.K.); milena.rizzo@ifc.cnr.it (M.R.); balzan@ifc.cnr.it (S.B.); iervasi@ifc.cnr.it (G.I.)

**Keywords:** mitoK-ATP, mitoK, mitoSur, low T3 state, miRNA, cardiac ischemia/reperfusion, Dio3

## Abstract

The opening of the ATP-sensitive mitochondrial potassium channel (mitok-ATP) is a common goal of cardioprotective strategies in the setting of acute and chronic myocardial disease. The biologically active thyroid hormone (TH), 3-5-3-triiodothyronine (T3), has been indicated as a potential activator of mitoK-ATP but the underlying mechanisms are still elusive. Here we describe a novel role of T3 in the transcriptional regulation of mitoK and mitoSur, the recently identified molecular constituents of the channel. To mimic human ischemic heart damage, we used a rat model of a low T3 state as the outcome of a myocardial ischemia/reperfusion event, and neonatal rat cardiomyocytes (NRCM) challenged with hypoxia or H_2_O_2_. Either in the in vivo or in vitro models, T3 administration to recover the physiological concentrations was able to restore the expression level of both the channel subunits, which were found to be downregulated under the stress conditions. Furthermore, the T3-mediated transcriptional activation of mitoK-ATP in the myocardium and NRCM was associated with the repression of the TH-inactivating enzyme, deiodinase 3 (Dio3), and an up-regulation of the T3-responsive miR-133a-3p. Mechanistically, the loss and gain of function experiments and reporter gene assays performed in NRCM, have revealed a new regulatory axis whereby the silencing of Dio3 under the control of miR-133a-3p drives the T3-dependent modulation of cardiac mitoK and mitoSur transcription.

## 1. Introduction

Cardiovascular disease is the leading cause of death worldwide [1]. In particular, acute myocardial infarction, evolving into post-ischemia/reperfusion (IR) heart disease, is the main culprit of myocardial damage. Though timely reperfusion has greatly improved the short-term outcome, it also produces further damage, including cardiomyocyte dysfunction and death, which in the long run favors an adverse remodeling of the heart chambers [2]. Consequently, the prevention or limitation of myocardial injuries in the early stages of reperfusion is a crucial step for improving the patient’s prognosis.

Multiple lines of evidence indicate mitochondrial dysfunctions as critical determinants of cell death during the acute phase of cardiac ischemia, as well as the progressive loss of surviving myocytes during the subacute and chronic phases [3,4,5]. Despite the encouraging mitochondria-targeted therapeutic strategies emerging from preclinical studies, very few have successfully completed clinical trials. Therefore, the mitochondrion remains a potentially untapped target for new therapies.

The opening of the ATP-sensitive mitochondrial potassium channel (mitoK-ATP) plays a pivotal antioxidant and protective role against post-IR mitochondrial dysfunction and cell loss [6,7]. In line with this notion, mitoK-ATP is the final effector of several cardioprotective approaches in both acute and chronic heart disease in several animal models [8,9,10].

Pioneering in vitro studies have explored the role of 3,5,3′ triiodothyronine (T3), the biologically active TH, in promoting the mitoK-ATP opening [11] but the underlying mechanisms are still unexplored.

THs are key regulators of the physiology of the mitochondrion [12] that represents the final effector of a large part of the T3-dependent cardioprotective action. Our previous investigations evidenced that about 30–40% of rats subjected to myocardial IR develop a condition similar to the low T3 state (LT3S) observed in patients after AMI, i.e., a reduction in serum T3 in the absence of overt thyroid disease. A timely (24 h after IR) and short-term (48 h) treatment with a physiological or near-physiological dose of T3 improved the post-ischemic recovery of cardiac function at 3 and 14 days [13,14,15]. In particular, correcting the low T3 state evoked many of the antioxidant and cytoprotective effects induced by mitoK-ATP activation [13,14,15].

The recent identification of mitoK and mitoSur as the constituent subunits of the mitoK-ATP channel has paved the way for studies aimed at a better understanding of the molecular mechanisms responsible for mitoK-ATP modulation [16]. Therefore, the main objective of the study was to analyze the involvement of T3 in the regulation of mitoK-ATP gene expression after acute myocardial IR injury.

For this purpose, we exploited both in vivo and in vitro experimental models of cardiac ischemic damage together with in silico prediction and functional analyses. Our results highlight, for the first time, a T3-driven regulatory circuit that fosters mitoK and mitoSur up-regulation facilitated by the miR-133a-dependent post-transcriptional inhibition of the TH inactivating enzyme iodothyronine deiodinase 3 (Dio3).

## 2. Results

### 2.1. T3 Replacement Is Associated with the Restoration of mitoK and mitoSur Expression Levels in Injured Cardiac Tissue and Cardiomyocytes

We first evaluated whether the post-IR beneficial effects of T3, which we previously observed in a myocardial IR rat model [13,14,15], were paralleled by changes in mitoK and mitoSur expression in the LV area at risk. To better mimic a physiological condition and from a translational perspective, we used a T3 dosage that was sufficient to correct the post-ischemic LT3S without inducing other systemic changes, i.e., a T3 replacement dose (3 μg/kg day) (Table 1).

As shown in Figure 1A, 3 days after IR in the untreated group, the LT3S was associated with a significant reduction in both mitoK and mitoSur transcript abundance. Correcting the LT3S by 48 h hormone infusion ensured the maintenance of Sham-like expression levels (Figure 1A). These data suggest that alterations to T3 homeostasis may influence mitoK and mitoSur production.

To strengthen such a connection, we then performed the expression analysis in a mouse model of hypothyroidism and cell cultures maintained in the presence or absence of T3. As shown in Figure 1B, hypothyroidism induced a significant down-regulation of cardiac mitoK and mitoSur with respect to the euthyroidism condition.

To mimic in vitro the in vivo post-IR cell damage, we used NRCM cells subjected to hypoxic or H_2_O_2_ stress, well-validated model systems for the analysis of cardiomyocyte injuries associated with ischemia [17,18]. In the in vitro models, 48 h incubation with T3 at a physiological concentration (3 nM) increased mitoK and mitoSur transcription both in basal conditions and after 6 h exposure to 300 μM of H_2_O_2_ (Figure 2A,B) or 24 h exposure to hypoxia (Figure 2D,E).

Of note, treatment with T3 was associated with a better-preserved mitoSur to mitoK ratio (Figure 2C,F), which may have relevant physiological implications. Indeed, it has recently been demonstrated that the presence of both subunits in well-defined proportions is necessary for the correct functioning of the mitoK-ATP channel [16]. It is therefore conceivable that the maintenance of an adequate mitoSur to mitoK ratio following T3 treatment plays a cytoprotective role in cardiomyocytes, especially under conditions of oxidative stress.

To confirm this hypothesis, we performed viability tests and assessed the inner mitochondrial membrane depolarization in NRCM treated for 6 h with 300 μM H_2_O_2_ and then 48 h with or without a physiological concentration of T3 (3 nM) in the presence or absence of 5-Hydroxydecanoic acid (5HD), a specific inhibitor of the mitoK-ATP channel (Figure 3A). Compared to the H_2_O_2_ group, T3-treated cells exhibited increased survival and better-preserved mitochondrial polarization, as evidenced by the increase in red absorbance in the Alamar blue test (Figure 3B) and red fluorescence in the JC1 test, respectively (Figure 3C). These effects were largely prevented by co-treatment with 5HD, confirming mitoK-ATP as a specific target for the protective action of T3 in cardiomyocytes.

Collectively the in vivo and in vitro data suggest a key role of TH fluctuation in determining the cardiac levels of mitoK and mitoSur, which may influence the activity of the channel.

### 2.2. The Cardiac Expression of Deiodinase 3 Is Inversely Associated with That of mitoK and mitoSur

To deepen our understanding of the mechanism of action of T3, we then performed an in silico analysis aimed at identifying putative binding sites for thyroid hormone receptors within the promoter of mitoK and mitoSur. Using the online transcription factor search tool “Promo”, we identified putative elements of recognition for the thyroid hormone receptor alpha (THRα) in the promoter region of both genes (*p* < 0.05) (see Appendix A).

Then, we reasoned that the stimulation of any factor capable of limiting the bioavailability of TH might switch off T3-responsive genes including mitoK and mitoSur. The TH-converting enzyme Dio3 is a fetal gene that converts thyroxin (T4) and T3 into their inactive counterparts. Therefore, we analyzed the expression levels of Dio3 in the IR rats and the two NRCM stress models. As reported in Figure 4A, the IR led to the up-regulation of Dio3 in the LV area at risk, which was prevented by LT3S correction. In NRCMs, treatment with T3 resulted in a significant down-regulation of the enzyme either in basal conditions or after the induction of cell stress (Figure 4B,C). By performing a linear regression analysis, we found that Dio3 correlated negatively with mitoK and mitoSur levels in IR rats (Figure 4D) and in stressed NRCM (Figure 4E,F).

These data support an increase in Dio3 production under stress conditions as a T3 modifiable phenomenon inversely associated with the expression of mitoK and mitoSur.

### 2.3. The T3-Dependent microRNA miR-133a Is Involved in the Activation of mitoK and mitoSur and the Post-Transcriptional Repression of Dio3

It is now widely accepted that T3-regulated microRNAs (miRNA) are involved in the silencing and activation of gene expression programs relevant to cardiac physiology and pathophysiology [19]. Among them, the conserved miR-133a-3p plays a critical cardioprotective role against post-ischemic fibrosis and mitochondrial dysfunction and is down-regulated by IR [13,14]. As we previously documented, the correction of the LT3S in the IR in vivo model is associated with the recovery of miR-133a-3p expression [13,14]. In accordance with these notions, here we confirmed a significant reduction of miR-133a-3p levels in NRCM under conditions of oxidative or hypoxic stress. Treatment with T3 counteracted the miR-133a repression observed in both stressful conditions (Figure 5A,B).

Given the shared mitoprotective effect of miR-133a and mitoK-ATP, we wondered whether miR-133a may play a role in the transcriptional modulation of mitoK and mitoSur. Thus, we analyzed the expression levels of the two-channel subunits in NRCM 48 h after transfection with a miRNA analog (miR-133a mimic) or its inhibitor (miR-133a decoy). As shown in Figure 5C,E, the miR-133a mimic caused an increase of about 50% in the transcript of both mitoK and mitoSur transcripts, which were instead repressed by about 40% in the presence of the miR-133a decoy (Figure 5E,F). These data indicate that miR-133a is involved in the transcriptional activation of mitoK and mitoSur and suggest an indirect effect of the miRNA on mitoK-ATP expression, probably mediated by the downregulation of an inhibitor or an inhibitory pathway.

Since the increase in miR-133a and Dio3 mRNA abundance results in opposite changes of mitoK and mitoSur transcription, we hypothesized that miR-133a may indirectly favor mitoK-ATP expression through Dio3 direct targeting. This hypothesis was initially confirmed by an in silico analysis using online miRNA target prediction tools. As reported in Figure 6A, the 3′ untranslated regions (3′UTR) of human, mouse, and rat Dio3 harbors recognition sites for miR-133a according to miRWalk and/or TargetScan predictive algorithms.

To validate the prediction, luciferase reporter assays were performed in NRCM. The 3′UTR of Dio3 was cloned downstream of the pGLU luciferase reporter constructs [20] and transfected into neonatal rat cardiomyocytes in the presence of increasing concentrations of the miR-133a mimic from 0 to 100 nM. The test, carried out 48 h after transfection, showed a progressive reduction in luciferase activity as the quantity of transfected miRNA increased, demonstrating that miR-133a directly binds the 3′UTR of Dio3 and inhibits translation (Figure 6B). This regulatory effect of miR-133a was further confirmed by analyzing the gene expression variations of Dio3 in NRCM transfected with the miR-133a mimic or miR-133a decoy. As reported in Figure 6C, the presence of the mimic and decoy resulted, respectively, in a significant reduction and increase in Dio3 expression levels, supporting the critical role of miR-133a in the post-transcriptional regulation of the gene.

## 3. Discussion

This study describes for the first time a key role of T3 in the transcriptional regulation of mitoK-ATP in cardiac tissue. The data obtained are in line with two possible interconnected mechanisms of action: (1) a direct activation of the gene expression of mitoK and mitoSur possibly mediated by THRα; and (2) an indirect derepressive effect due to the miR-133a-guided inhibition of the TH inactivating enzyme Dio3.

In the myocardium of IR rats, the early recovery of euthyroidism is accompanied by the maintenance of the post-ischemic levels of mitoK and mitoSur transcripts, which is in line with the previously reported improvement in both short-term (3 d) and medium-term (14 d) morpho-functional parameters including LV end-systolic diameter, ejection fraction, and fractional shortening, and resulted in the long-lasting normalization of cardiac performance and geometry [13,14,15]. In agreement, the pharmacological inhibition of mitoK-ATP largely prevented the protective effect of T3 against NRCM mitochondrial dysfunctions and death in the presence of oxidative stress. Notably, in both the in vitro models of post-ischemic injuries, T3 ensures the maintenance of an adequate mitoSur/mitoK ratio, which, in light of recent evidence, is an indispensable requirement for guaranteeing the full functionality of the channel [16].

On the other hand, managing oxidative stress damages in the early stages of IR is a primary goal of cardioprotection to limit the loss of cardiomyocytes and blunt adverse cardiac remodeling. In the post-IR setting, mitochondria are both the primary source and main targets of reactive oxygen species (ROS). In turn, mitochondrial dysfunctions triggered by ROS and the accumulation of calcium in the mitochondrial matrix favor cell death and inflammation [17]. The activation of mitoK-ATP under stress conditions has been proposed to cause slight mitochondrial uncoupling, which prevents ROS generation through the reverse electron transport mechanism and limits calcium overload [6,7,21]. Therefore, we propose that T3 replacement favors the antioxidant activity of mitok-ATP by upregulating the subunits of the channel under stress conditions. Accordingly, mitoK-ATP is the final effector of cardioprotective strategies in acute IR injury and in preclinical models of diabetic and hypertrophic cardiomyopathy [8,9,10,22]. Moreover, in arrhythmic patients, a significant reduction in mitoK-ATP expression is associated with an increased atrial fibrillation rate [23].

In this scenario, our data point to the mitochondrial channel as a pivotal mediator of T3 cardiac activity. Although suggestions toward this concept have already emerged in aprevious work [11], the recent identification of the molecular constituents of mitoK-ATP made it possible to carry out a more in-depth investigation of the potential underlying mechanisms. The in silico analysis has in fact highlighted the presence of consensus sites for THRα in the promoters of mitoK and mitoSur. Even though we did not perform an in vitro functional validation, which will be the subject of future studies, these findings are in line with the well-known T3-dependent transcriptional modulation of several ion channels involved in electromechanical coupling [24]. The reliability of the prediction is further supported by the fact that the algorithm has identified the consensus sequences for the THRα, which represents the most abundant isoform in the heart and is mainly involved in the cardioprotective action of T3 [25]. In light of the well-known bimodal switch model of THR-mediated chromatin remodeling, it can be speculated that in the presence of a cardiac LT3S, the unbound THRα acts as a repressor of mitoK and mitoSur. This interpretation is supported by the results obtained in the hypothyroidism model in which the cardiac levels of the channel subunits are significantly reduced compared to euthyroid animals.

The LT3S we observed in our mode must be framed in the broader perspective of a reactivation of the fetal gene expression program, probably supported by the induction of Dio3. According to this paradigm, in the adult heart, disease signals can reactivate Dio3 expression leading to a Dio3-mediated decrease in T3 signaling, which ultimately slows down high-energy-consuming adult protein isoforms in favor of the fetal ones. An up-regulation of Dio3 activity as a possible cause of a local condition of hypothyroidism in the decompensated heart, was first observed in a mouse model of right ventricle pathological hypertrophy induced by pulmonary arterial hypertension [26]. The ventricular-specific expression of Dio3 correlated with a gene expression profile characteristic of pathological remodeling and was associated with a reduction in tissue levels of T3 [27,28]. Since then, the induction of the cardiac activity of Dio3 has been found in models of pathological remodeling of the LV induced by aortic stenosis, chronic myocardial infarction, isoproterenol, and diabetes mellitus [29,30,31,32]. To the best of our knowledge, the present study is the first to indicate a significant induction of the TH-inactivating enzyme in the early stages of the IR process. The restoration of physiological T3 concentrations resulted in the silencing of Dio3 both in the rat IR model and in NRCM subjected to hypoxic and oxidative stress. It is plausible that, similar to the postnatal period of cardiac differentiation, in the post-IR conditions the expression of Dio3 is silenced as the levels of T3 increase.

Strictly connected with the differentiation/de-differentiation processes, our results show, for the first time, a fundamental role of the T3-sensitive miR-133a in the post-transcriptional inhibition of Dio3. MiR-133a is considered a myomiRNA as it shows a specific expression in cardiac and skeletal muscle where it exerts a pro-differentiative activity. During the early stages of cardiac development, miR-133a directs the maturation of mesodermal cells toward the cardio-specific muscle lineage; furthermore, it shows significant cardiogenic potential in the reprogramming of fibroblasts into cardiomyocytes [33]. MiR-133a is also involved in cardiac remodeling in response to various stressful stimuli [34]. The myocardial down-regulation of miR-133a correlates with disease severity in patients with heart failure, and with increased incidence of fatal ventricular arrhythmias in patients with acute myocardial infarction [35,36]. In various animal models of post-ischemic heart disease, miR-133a exerts multiple protective functions related to the regulation of apoptosis, cardiac fibrosis, cardiac hypertrophy, and electrical activities [37].

Consistent with these clinical and experimental data, a reduced expression of miR-133a in the LV area at risk was previously observed in our IR model [13,14]. In those studies, the miRNA deregulation, found in the early stages of post-IR damage, was prevented by treatment with T3 at physiological concentrations. Here, those observations have been reproduced in in vitro models of cardiac stress in which T3 limited cell death and mitochondrial depolarization in a mitoK-ATP-dependent manner. Overall, our in vitro results suggest that maintenance of miR-133a levels and activation of mitoK-ATP may be interconnected mechanisms of the T3 cardioprotective action. In favor of such a connection, miR-133a is important for the transcriptional activation of mitoK and mitoSur in NRCM culture. In the same direction, the up-regulation of miR-133a in an animal model of myocardial ischemic post-conditioning has been reported to exert a protective effect similar to mitoK-ATP activation [38].

Overall, the data discussed so far, lead us to hypothesize a model whereby the noxious environment induced by post-ischemic reperfusion reactivates the gene program of the immature heart (Figure 7). In this context, the increase in the fetal gene Dio3 lowers the transcript levels of mitoK, mitoSur, and mirR-133a by reducing the local bioavailability of T3. In turn, the cardiac repression of miR-133a contributes to the further overexpression of Dio3 through a mechanism of gene derepression. In this scenario, the restoration of euthyroid hormone levels by exogenous administration of T3 at physiological concentrations produces a normalization of the cardiac levels of miR-133a, re-establishing the negative regulation of Dio3 expression and, therefore, allowing the transcriptional reactivation of the T3-sensitive genes mitoK and mitoSur.

However, it cannot be excluded that other pathways evoked by the T3-sensitive mir-133a might influence the expression of mitoK and mitoSur.

In conclusion, our findings point to the post-ischemic LT3S as a permissive condition for the inhibition of mitoK and mitoSur. An approach aimed at restoring the plasma and myocardial levels of T3 could represent a favorable strategy to limit the evolution of post-ischemic heart disease. To be effective against adverse remodeling, we propose a timely reconstitution of euthyroidism in the early phase of the post-IR wound-healing process, a time when the activation of harmful mitochondrial regulatory networks has not yet produced permanent effects and can be prevented or mitigated, at least in part, by de-repression of the mitoK-ATP channel.

## 4. Materials and Methods

### 4.1. Experimental Design and Animal Models

The experimental design included an in vivo exploratory investigation of established rat models of acute myocardial ischemia and reperfusion (IR) and hypothyroidism, followed by an in vitro validation study on neonatal rat cardiomyocytes (NRCM). The left ventricle samples used in the in vivo study were collected during previous works [13,14,39]. All surgery was performed under anesthesia and all efforts were made to minimize suffering. 

Briefly, myocardial infarction was produced by ligation of the left descending coronary artery of 12–15 week-old adult male *Wistar* rats (Charles River Wilmington, MA, USA). After 30 min of ischemia, unrestrained reperfusion was allowed for 3 d. A control group of rats underwent all surgical procedures except for the occlusion of the LAD (Sham group). To mimic the typical low T3 state (LT3S) of the post-IR clinical setting, only the IR rats which exhibited a >50% reduction in their basal serum FT3 level 24 h after surgery were randomly treated for 48  h with a constant subcutaneous infusion of 3  μg/kg/day T3 (IRT3 group) or saline (IR group) via a mini osmotic pump (Alzet, model 2ML4, Palo Alto, CA, USA).

Hypothyroidism was produced in *Wistar* rats (2–3 months old) (Charles River Wilmington, MA, USA) by administration of 6-propyl-2-thiouracil (PTU) in drinking water at a final concentration of 0.05% (weight/volume) for 3 weeks (hypothyroid group, IPO) or its vehicle for the same time (euthyroid group; EU) as previously described [39]. In the PTU group, the condition of hypothyroidism was considered reached for serum levels of free FT3 < 1.5 pmol/L against control values of 4.5 pml/L and for not quantifiable FT4 level against values of 13.7 pmol/L in the control group [39].

At the end of each specific treatment, the animals were anesthetized to take 2 mL of blood from the femoral vein for serum thyroid hormone assay and then sacrificed by lethal injection of KCl. The heart was removed to collect samples from the area at risk or corresponding LV region in the rats not subjected to IR. Tissue was immediately frozen in liquid nitrogen and stored at −80 °C until the moment of use. The heart tissue was immediately frozen in liquid nitrogen and stored at −80 °C. To avoid repeated freeze/thawing cycles, at the first use the frozen tissue was pulverized in liquid nitrogen and subdivided into ready-to-use aliquots stored at −80 °C

The level of serum-free T3 (FT3) and T4 (FT4) were assayed as previously described [39,40].

### 4.2. Isolation of Neonatal Rat Cardiomyocytes 

NRCM were isolated through enzymatic digestion from hearts of 2 to 3 day-old *Wistar* rats using the Worthington Neonatal Cardiomyocyte Isolation System (Worthington, Lakewood, NJ, USA) according to the manufacturer’s instructions and as previously described [41]. To prevent the overgrowth of contaminating fibroblasts, a 2 h pre-plating procedure was adopted prior to NRCM plating and 10 uM Bromodeoxyuridine was added to the growing medium. NRCM were grown in a humidified atmosphere of 5% CO_2_ at 37 °C in DMEM low glucose with 2 mM l-glutamine, 1% penicillin and streptomycin, and 10% fetal bovine serum (FBS, Merck, Darmstadt, Germany) (complete medium). In the experiments involving T3 treatment (see below), FBS was substituted with the same amount of FBS deprived of TH with standard charcoal stripping procedures (TH-free medium). Briefly, FBS was absorbed overnight at 4 °C on dextran-coated activated charcoal (40 μg/mL serum), the suspension was centrifuged to precipitate the charcoal and the supernatant was filtered through 0.2 μm filters.

### 4.3. In Vitro Stress Protocols

For the hypoxic stress, 250,000 NRCM were plated in 3 cm Petri dishes (Sarstedt) in a complete medium. After plating for 24 h, the medium was replaced with L15 containing 2% serum and the plates were placed in a hypoxia chamber (STEMCELL Technologies, Köln, Germany) and exposed to a 20 L/min nitrogen flow for 10 min to remove oxygen. The hermetically sealed chamber was placed in the incubator for 24 h. Subsequently, the medium was changed and replaced with DMEM containing 3 nM of T3 (HypoxT3 group) or its vehicle for 48 h (Hypox group). Unstressed cells maintained in 3 nM T3 (T3 group) or T3 vehicle (Control group) served as control.

For the oxidative stress, 24 h after plating NRCM were exposed to 6h treatment with H_2_O_2_ (300 μM) (Merck, Darmstadt, Germany) in low-glucose DMEM (Merck, Darmstadt, Germany) containing 2% FBS. Thereafter, the medium was changed and NRCM were maintained for a further 48 h with a complete medium containing the following treatment: 3 nM T3 (H_2_O_2_-T3 group) or its vehicle (H_2_O_2_ group), 500 μM of the specific mitok-ATP inhibitor 5-hydroxy decanoic acid alone (H_2_O_2_-5HD group), or with 3 nM T3 (H_2_O_2_-5HD-T3 group). Unstressed cells maintained in 3 nM T3 (T3 group) or T3 vehicle (Control group) served as control.

### 4.4. Viability Assay

For the viability test, 10,000 CM per well were plated in a 96-multiwall plate. At the end of the experimental protocol described above and shown in Figure 3A, the Alamar blue test (Thermo Fisher, Waltham, MA, USA) was performed according to the manufacturer’s instructions. Briefly, the cells were incubated with 200 µL of a 5% solution of Alamar blue reagent diluted in a culture medium. After 4–6 h of incubation at 37 °C, the absorbance at 570 nm and 600 nm was read using the Infinite M200 Pro (Tecan, Männedorf, Switzerland) multifunction plate reader and the Icontrol software (Tecan, Männedorf, Switzerland).

### 4.5. Measurement of the Inner Mitochondrial Membrane Potential

Alterations to the inner mitochondrial membrane potential (Δψm), an indicator of cellular and mitochondrial damage, were evaluated using the mitochondria-targeted fluorescent probe 5,5′,6,6′-tetrachloro-1,1′,3,3′-tetraethyl-imidacarbocyanine iodide (JC-1) (Merck, Darmstadt, Germany). JC-1 accumulates in mitochondria in a membrane-potential-dependent manner, as indicated by a fluorescence emission shift from green (JC-1 monomers) to red (JC-1 aggregates). Therefore, a reduction in the fluorescence value of the aggregates is indicative of depolarization, whereas an increase is indicative of hyperpolarization. The conditions and experimental groups used for the test with JC1 are illustrated in Figure 3A. At the end of the experimental protocol, the NRCMs were incubated with JC-1 10 µM for 15 min at 37 °C in the dark. The aggregates of JC-1 (excitation wavelength 525 nm; emission wavelength 590 nm) and the monomers of JC-1 (excitation wavelength 490 nm; emission 530 nm) were measured using the Infinite M200 Pro (Tecan, Männedorf, Switzerland) plate reader and Icontrol (Tecan, Männedorf, Switzerland) software.

### 4.6. In Silico Analyses

Putative thyroid hormone receptor consensus binding sites on the promoter of rat mitoK (also known as CCDC51) and mitoSur (also known as ABCB8) were identified through the online search tool “Promo” available at the following link: “http://alggen.lsi.upc.es/cgibin/promo_v3/promo/promoinit.cgi?dirDB=TF_6.4 (accessed on 25 May 2022)”. For the research, the 5000 nucleotides upstream of the transcription starting site were used as input, setting *p* < 0.05 and the maximum rate of dissimilarity of the matrix at 15% (See Appendix A). The promoter regions of mitoK and mitoSur were extrapolated from NCBI (National Center for Biotechnology Information) and EPD (Eukaryotic Promoter Database) available at the following links: “https://www.ncbi.nlm.nih.gov/ (accessed on 25 May 2022)” and “https://epd.epfl.ch//index.php (accessed on 25 May 2022)”.

The prediction of the interaction between miR-133a-3p and the 3′UTR of human, rat, and mouse Dio3 transcripts was performed using the online tools miRWalk and TargeSscan available at the following links:

“http://mirwalk.umm.uni-heidelberg.de/ (accessed on 25 May 2022)” and “http://www.targetscan.org/vert_80/ (accessed on 25 May 2022)”.

### 4.7. Overexpression and Downregulation of miR-133a-3p in NRCM

For miR-133a overexpression or repression experiments, NRCM were plated in 12-multiwall plates at a density of approximately 150,000 cells per well. After plating for 24 h, the CMs were transfected for 5 h with a mixture containing 1.5 µL of transfectant lipofectamine 2000 (Invitrogen, Waltham, MA, USA) and 100 nM of the miR-133a mimic or control mimic (Eurofins Genomics, Ebersberg, Germany), or 2′O-methylated-mir133a-decoy or control 2′O-methylated decoy (Eurofins Genomics, Ebersberg, Germany). The mimic and decoy sequences are shown in Appendix A. At the end of the transfection protocol, the cells were kept for 48 h in a complete medium and then used in the gene expression analysis.

### 4.8. Plasmid Construction and Reporter Assays

The dual firefly-and-renilla/luciferase-reporter gene assay was performed in order to evaluate the ability of miR-133a-3p to directly bind Dio3 3′UTR and repress translation. The reporter vector was generated by cloning the entire sequence of the 3′ untranslated regions (3′UTR) of the mouse Dio3 transcript (Appendix A) downstream of the luciferase reporter gene of the pGLU Dual-luciferase reporter plasmid vector [20].

The sequence of interest was amplified by PCR using the Phusion™ High-Fidelity DNA Polymerase (ThermoFisher, Waltham, MA, USA). After purification through gel-electrophoresis and extraction with the Qiaquick gel extraction kit (Qiagen, Hilden, Germany), the 3′UTR was cloned into the pGEM-T Easy vector (Promega, Madison, WI, USA) and sequenced to confirm that no errors were introduced during the procedures. Next, the insert was excised by enzymatic digestion with KpnI and Nhe1 and sub-cloned into the KpnI/Nhe1 sites within the polylinker region of the pGLU Dual-luciferase vector downstream of the luciferase gene. The screening of the colonies containing the correct insert, and the sequencing and construct extraction were performed as described above for the pGEM-T Easy vector.

For the luciferase assays, 150,000 CM/well were plated in 12-well plates. After plating for 24 h, the CMs were cotransfected via 6 h incubation with 500 ng of reporter vector, 100 ng of pRL control vector-TK (Renilla, Promega, Madison, WI, USA), and increasing concentrations of the miR-133a mimic (0.25 nM, 50 nM, 100 nM). The control mimic was used to ensure an equal total concentration of the miR mimic in each well. As transfectant, 1.5 µL aliquots of lipofectamine 2000/well (Invitrogen) were used according to the manufacturer’s instructions. After transfection for 48 h, the cells were processed with the Dual-Luciferase^®^ Reporter Assay System (Promega, Madison, WI, USA) kit according to the manufacturer’s instructions. Briefly, after cell lysis and recovery of the cytosolic fraction by 10 min centrifugation at 12,000 rpm, the luminescence was quantified in 20 µL of the sample using the GloMax-Multi detection system (Promega, Madison, WI, USA) luminometer.

### 4.9. RNA Extraction and Real-Time PCR

Total RNA was extracted from tissue samples and NRCM with the miRNeasy mini kit reagent (Qiagen, Hilden, Germany) according to the manufacturer’s instructions. RNA quality and amount were determined using the Agilent Bioanalyzer 2100 and the RNA 6000 Nano Kit (Agilent Technologies, Santa Clara, CA, USA) The cDNA was synthesized from 1 μg RNA using the QuantiTect Reverse Transcription Kit (Qiagen, Hilden, Germany) or miScript II RT kit (Qiagen, Hilden, Germany) as indicated by the manufacturer. 

For gene and miRNA expression analyses, 10 ng of cDNA were processed in triplicate in a Rotor-Gene Q real-time machine (Qiagen, Hilden, Germany) using the Quantifast SYBR Green Mix (Qiagen, Hilden, Germany). PCR conditions were as follows: 5 min of initial denaturation and then 40 cycles of 95 °C for 10 s, 58 °C for 20 s, and 72 °C for 10 s. To assess product specificity, a melting curve analysis from 65 °C to 95 °C with a heating rate of 0.1 °C/s with a continuous fluorescence acquisition was constructed. Gene transcript values were normalized using Hprt and Hmbs reference genes. miRNA transcript values were normalized using U6 and U1 reference miRNA. The relative quantification of samples was performed by Rotor Gene Q-Series Software (Qiagen, Hilden, Germany) and expressed as mean ± standard error of the mean (SEM). The complete list of primer sequences is shown in Appendix A.

### 4.10. Statistical Analysis

All variables met the condition for parametric analysis. The differences between the means of two variables were evaluated using the Student’s *t*-test. Comparisons between more than two groups were performed using one-way ANOVA, followed by Bonferroni’s post hoc correction (IBM Spss 20 statistic, Armonk, NY, USA). Linear regression analysis was performed using the IBM Spss 20 statistics software. Results are expressed as mean ± SEM and *p* values < 0.05 were considered statistically significant. 

## Figures and Tables

**Figure 1 ijms-23-06549-f001:**
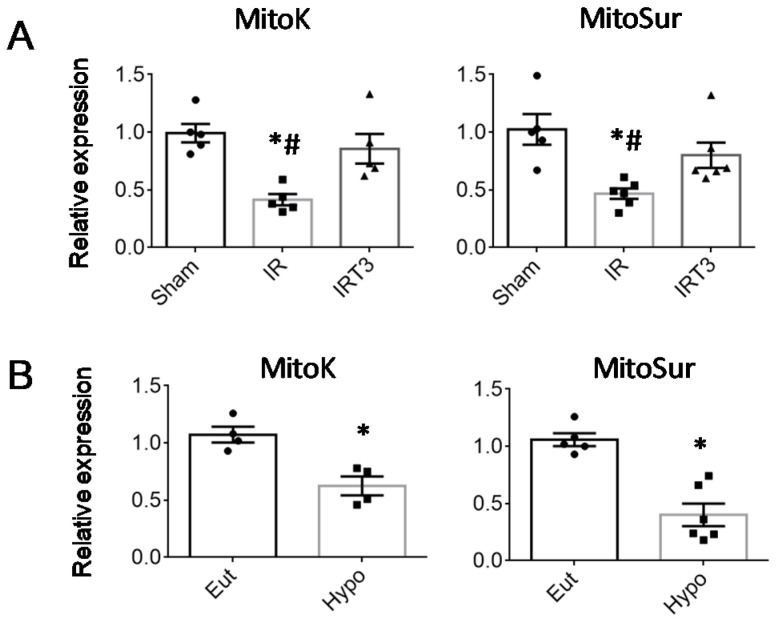
T3 restores mitoK and mitoSur expression levels in the rat heart. The relative expression of mitoK and mitoSur was measured (**A**) in the LV area at risk at 3 d from the IR procedure followed by treatment with T3 at physiological doses (3 μg per kg per day) for 48 h (IRT3). *n* = 5 animals per group, * *p* = 0.002 vs. Sham; # *p* ≤ 0.02 vs. IRT3; and (**B**) in LV from euthyroid (Eut) and hypothyroid (Hypo) rats. *n* = 4 animals per group, * *p* < 0.0001 vs. Eut.

**Figure 2 ijms-23-06549-f002:**
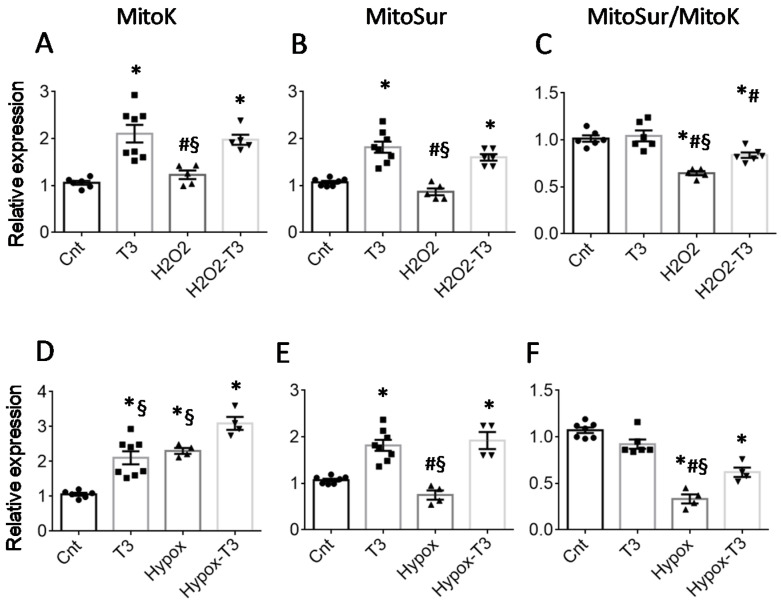
Treatment with T3 induces the expression of mitoK and mitoSur in cultured neonatal rat cardiomyocytes. Upper panels (**A**–**C**): relative expression of mitoK and mitoSur in NRCM stressed for 6 h with 300 μM H_2_O_2_ and then treated for 48 h with 3 nM T3 or its vehicle. *n* ≥ 5, * *p* < 0.03 vs. Cnt; # *p* ≤ 0.03 vs. T3; § *p* ≤ 0.04 vs. H_2_O_2_T3. Lower panels (**D**–**F**): relative expression of mitoK and mitoSur in rat cardiomyocyte cell cultures stressed for 24 h with hypoxia and then treated for 48 h with 3 nM T3 or its vehicle. *n* ≥ 4, * *p* ≤ 0.002 vs. Cnt; # *p* <0.0001 vs. T3; § *p* < 0.03 vs. Hypox-T3.

**Figure 3 ijms-23-06549-f003:**
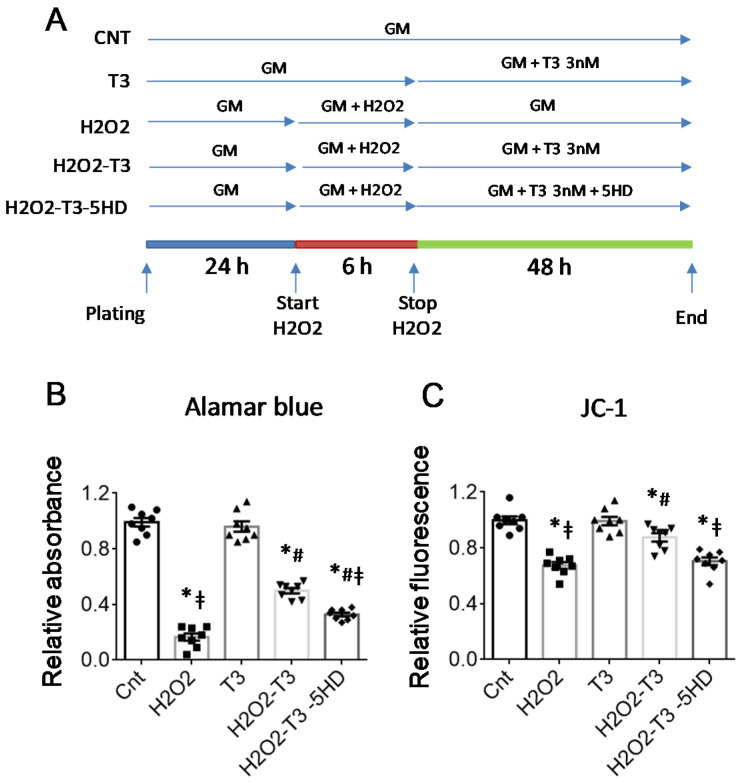
The pharmacological inhibition of mitoK-ATP reduces the protective effect of T3 against cell death and alterations to the inner mitochondrial membrane potential. (**A**) Schematic representation of the experimental protocol. GM = growth medium, 5HD = 5 -Hydroxydecanoate. (**B**) Viability test; and (**C**) JC1 assay in NRCM stressed for 6 h with 300 μM H_2_O_2_ and then treated with 3 nM T3 or its vehicle for 48 h. *n* = 8, * *p* < 0.0001 vs. Cnt and T3; # *p* ≤ 0.001 vs. H_2_O_2_; ‡ *p* ≤ 0.02 vs. H2O2-T3.

**Figure 4 ijms-23-06549-f004:**
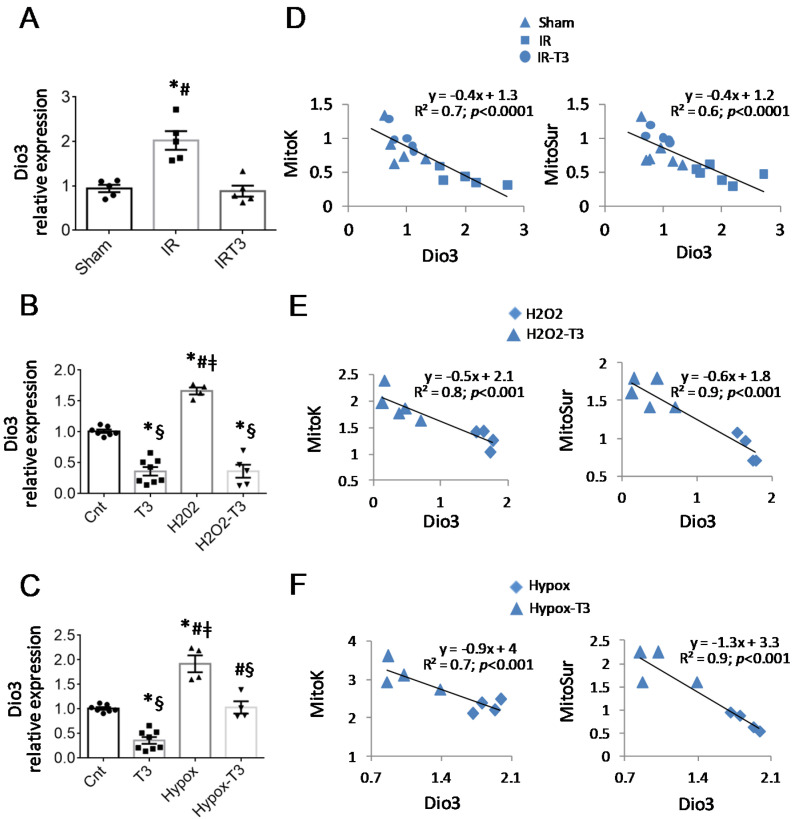
Treatment with T3 reduces the expression of Dio3 in in vivo and in vitro pathological conditions. (**A**) Quantification of the relative expression of Dio3 in the LV area at risk 3 days after IR induction (*n* = 5 animals per group; * *p* < 0.0001 vs. sham; # *p* < 0.0001 vs. IRT3. (**B**,**C**) Quantification of the relative expression of Dio3 in NRCM subjected to hypoxic or oxidative stress and then treated with T3 or its vehicle for 48 h. *n* ≥ 4; * *p* < 0.0001 vs. cnt; # *p* < 0.0001 vs. T3; § *p* < 0.0001 vs. Ipox or H_2_O_2_; ‡ *p* < 0.0001 vs. Ipox-T3 or H2O2-T3. (**D**) Linear regression analysis between expression levels of Dio3 (independent variable) and mitoK or mitoSur (dependent variables) in the LV area at risk at 3d post IR. (**E**,**F**) Linear regression analysis between expression levels of Dio3 (independent variable) and mitoK or mitoSur (dependent variables) in NRCM subjected to H_2_O_2_ or hypoxic stress.

**Figure 5 ijms-23-06549-f005:**
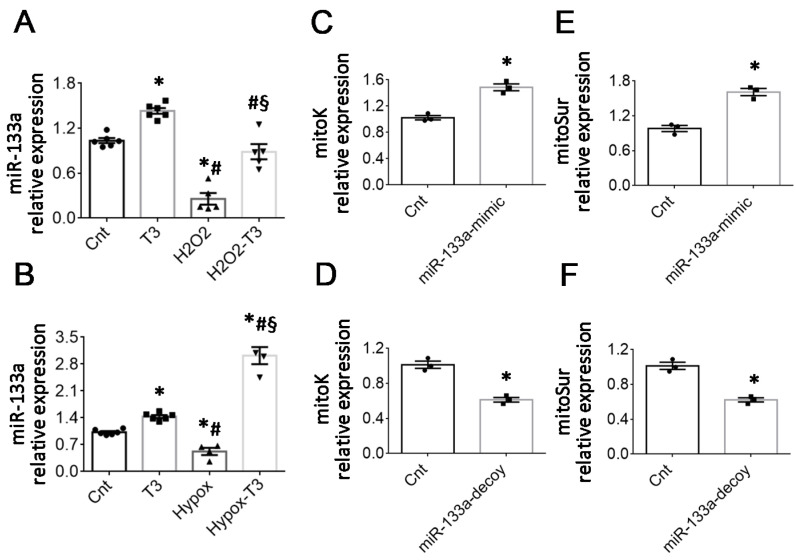
The T3-responsive miR-133a is involved in the up-regulation of mitoK and mitoSur. (**A**,**B**) Relative expression of miR-133a in the two in vitro models of oxidative and hypoxic stress. *n* ≥ 4, * *p* ≤ 0.03 vs. Cnt and T3; # *p* < 0.0001 vs. T3; § *p* < 0.0001 vs. Hypox or H_2_O_2_. (**C**–**F**) Relative expression of mitoK (**C**,**D**) and mitosur (**E**,**F**) in cultured neonatal cardiomyocytes 48 h post-transfection with miR-133a mimic or miR-133 decoy. *n* = 3 * *p* ≤ 0.001 vs. the respective Cnt.

**Figure 6 ijms-23-06549-f006:**
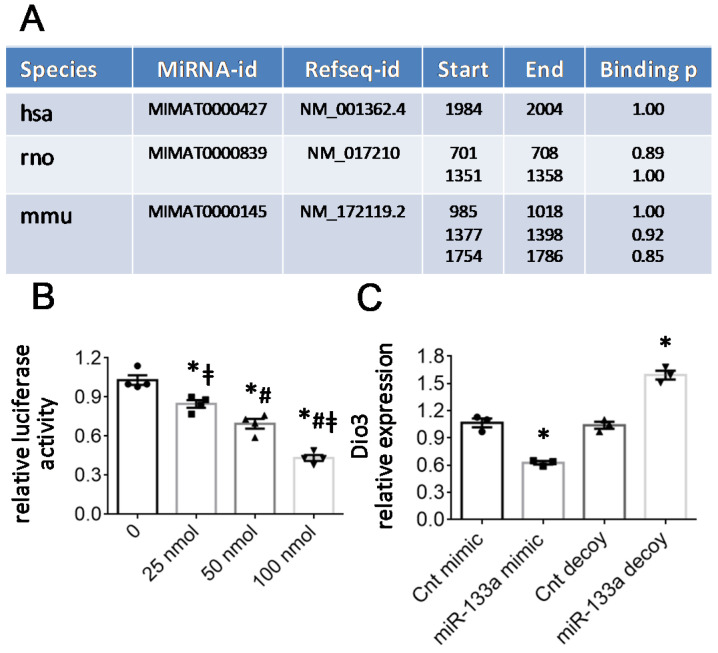
Dio3 transcript is a direct target of the T3-sensitive miR-133a. (**A**) Predicted pairing between the 3′UTR of the human, rat, and mouse Dio3 and the seed sequence of the conserved miR-133a-3p as reported by the miRWalk and/or TargetScan predictive algorithms. (**B**) The entire mmu 3′UTR of Dio3 has been cloned in a reporter vector downstream of the luciferase gene and transfected in neonatal rat cardiomyocytes. The co-transfection with miR-133a reduced luciferase activity in a dose-dependent manner. *n* = 4, * *p* ≤ 0.004 vs. Cnt; # *p* ≤ 0.01 vs. 25 nM; ‡ *p* ≤ 0.01 vs. 50 nM. (**C**) Relative expression of Dio3 in cultured neonatal cardiomyocytes 48 h after transfection with miR-133a mimic or miR-133 decoy. *n* = 3, * *p* ≤ 0.007 vs. respective Cnt.

**Figure 7 ijms-23-06549-f007:**
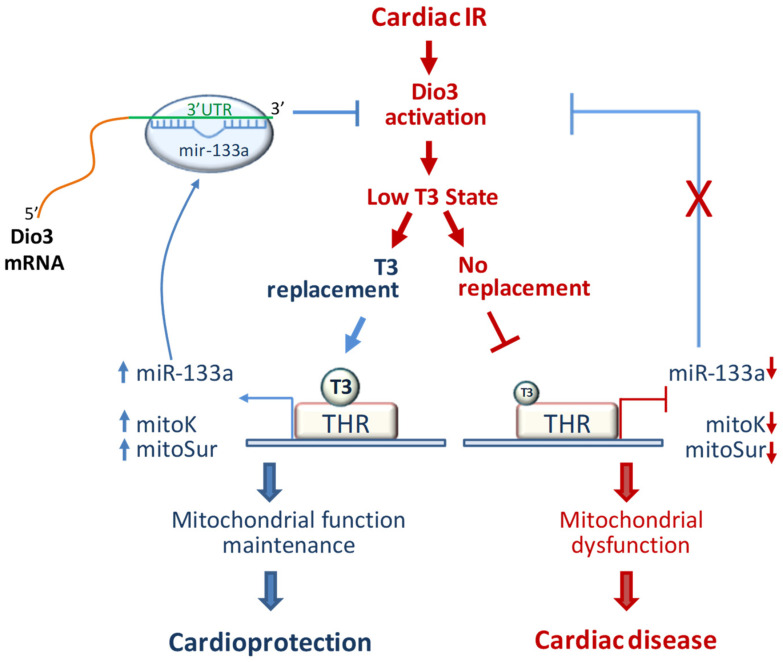
Schematic summary of the action of T3 through the miR-133a-Dio3 axis in the post-IR myocardium. Adverse pathways induced by ischemic stress and cardiac reperfusion are shown in red, whereas the cardioprotective pathways activated by the exogenous administration of T3 are represented in blue. THR = thyroid hormone receptor.

**Table 1 ijms-23-06549-t001:** Plasma level of free T3 and T4 at 3 d post-surgery.

	Sham	IR	IRT3
FT3 (pg/mL)	3.0 ± 0.2	2.0 ± 0.01 *#	2.9 ± 0.06
FT4 (pg/mL)	11.9 ± 1.05	12.6 ± 1.35	10.4 ± 1.5

*n*  =  5 animals per group * *p*  <  0.01 vs. Sham and # *p* <  0.01 vs. IRT3.

## Data Availability

All data generated or analyzed during this study are included in this article (and its Appendix A).

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
