# Peer review of "Role of miR-133/Dio3 Axis in the T3-Dependent Modulation of Cardiac mitoK-ATP Expression"

_ijms, 2022, doi:10.3390/ijms23126549_

Round 1

Reviewer 1 Report

Comments in the attached file.

Author Response

Thanks to the authors of the article for such an interesting work. These articles are undoubtedly important and can be further applied to the treatment of various heart pathologies.

Replay: We have really appreciated the positive remarks of the reviewer and are grateful for his/her constructive comments and suggestions.

Comments:

1 Line 71. We firstly evaluated whether the post IR beneficial effects of T3, previously observed in our model. What is our model? Description or reference?

Replay: The model we refer to is the cardiac ischemia reperfusion model, we have briefly introduced it in the introduction section, lines 50-55 as follows “Our previous investigations evidenced that about 30–40% of rats subjected to myocardial IR develops a condition similar to the low T3 state (LT3S) observed in patients after AMI, i.e., a reduction of serum T3 in the absence of overt thyroid disease. A timely (24 h after IR) and short term (48h) treatment with physiological or near-physiological dose of T3 improved the post ischemic recovery of cardiac function at 3 and 14 days [13-15]”. Also, the model is described in the material and method section par. 4.1 lines 348-355. According to the reviewer’s suggestion a more explicit reference to the model has been reported in the result section, line71, as follows: “We firstly evaluated whether the post IR beneficial effects of T3, we previously observed in a myocardial IR rat model [13-15]…..”.

2 Has the activity of the mitoK channel itself been measured in mitochondria during ischemia/reperfusion in an animal model as well as mitochondria from NRCM cells during hypoxic and oxidative stress?

Replay: Unfortunately we are not equipped to assess channel activity. Though this evaluation should be of great interest, we have focused on  transcriptional regulation of the mitoK-ATP channel

3 Line 80. (a) in the LV area at risk at 3 from the IR procedure. Missing word days.

Replay: The manuscript has been modified accordingly as follows: “in the LV area at risk at 3d from…”

4 Line 330. The left ventricle samples used in the in vivo study were collected during previous works [13,14,40]. Does this mean that you used left ventricular rat samples that were taken and frozen at -80°C 4-6 years ago, judging by the links to the articles? Could you add how quickly the tissue samples were frozen after decapitation of the animals, storage conditions, were the tissues thawed in those 4-6 years, how many times?

Replay: We thank the reviewer for giving us the opportunity to clarify this important issue. Upon removal, the heart tissue was immediately frozen in liquid nitrogen and stored at-80°C. To avoid repeated freeze/thawing cycles, at the first use the frozen tissue was pulverized in liquid nitrogen and subdivided into ready-to-use aliquots stored at -80°C. This information has been added in the material and methods section as follows: “The heart tissue was immediately frozen in liquid nitrogen and stored at-80°C. To avoid repeated freeze/thawing cycles, at the first use the frozen tissue was pulverized in liquid nitrogen and subdivided into ready-to-use aliquots stored at -80°C”, Lines 366-370 Moreover, to exclude sample deterioration, we performed RNA quality check after each extraction. We add this information at lines 478-479 as follows: “RNA quality and amount was determined using the Agilent Bioanalyzer 2100 and the RNA 6000 Nano Kit (Agilent Technologies”.

5 Line 384 –blu Misprint?

Replay: Sorry, the typo has been corrected.

6 Line 71. We firstly evaluated whether the post IR beneficial effects of T3, previously observed in our model, were paralleled by changes of mitoK and mitoSur expression in the LV area at risk. What kind of effects are mentioned, what model is mentioned?

Replay: We hope to have now clarified this point, as indicated in the replay to comment 1.

7 Since excess T3 concentrations can lead to oxidative stress (doi.org/10.1016/j.mito.2020.04.005; doi: 10.3390/ijms222111744N and many others) and although you use physiological T3 concentrations when treating pathology, it would be good to add a figure/table with T4/T3 concentrations in Sham, IR, IRT groups to the results.

Replay: Accordingly to the reviewer’s suggestion, in the revised version we have added a new table (Table 1) reporting the plasma concentrations of free T3 (FT3) and free T4 (FT4). See lines 78-79. Moreover, a sentence has been added at lines 73-76 to betters explain  the concept of T3 replacement as follows: “To better mimic a physiological condition and in a translational perspective, we used a T3 dosage that was sufficient to correct the post ischemic LT3S without inducing other systemic changes, i.e. a T3 replacement dose (3 μg/Kg day) (Table 1).” Lines 73-76.

8 Line 84. To strengthen such connection, we then performed the expression analyzes in a mouse model of hypothyroidism and in cell cultures maintained in the presence or absence of T3. As shown in Figure 1b. There is no mention of a mouse model of hypothyroidism in the materials and methods. Figure 1b refers to rats. Characteristics of hypothyroidism in animals?

Replay: We apologize for the mistake, indeed the hypothyroid animals were rats, the text was corrected accordingly. A more detailed description of the Hypothyroidism model has been reported in the material and method section in lines 359-362 as follows: “In the PTU group the condition of hypothyroidism was considered for serum levels of free FT3 <1.5 pmol/L against control values of 4.5 pml /L and  for not quantifiable FT4 level against values of  13.7 pmol/L in the control group [40]

9 Did you use models of oxidative stress and hypoxia? How did you check the development of oxidative stress and signs of hypoxia?

Replay: In the case of H2O2 injury, the extent of stress has been evaluated indirectly as a reduction of cell viability and decrease of mitochondrial polarization. As far as the hypoxia model is concerned, the stress was confirmed by the downregulation of the alpha myosin heavy chain, and the upregulation of apelin (data not shown), two well established markers of hypoxia (Razeghi P, Essop MF, Huss JM, Abbasi S, Manga N, Taegtmeyer H. Hypoxia-induced switches of myosin heavy chain iso-gene expression in rat heart. Biochem Biophys Res Commun. 2003 Apr 18;303(4):1024-7. doi: 10.1016/s0006-291x(03)00478-9. PMID: 12684037; Ronkainen VP, Ronkainen JJ, Hänninen SL, Leskinen H, Ruas JL, Pereira T, Poellinger L, Vuolteenaho O, Tavi P. Hypoxia inducible factor regulates the cardiac expression and secretion of apelin. FASEB J. 2007 Jun;21(8):1821-30. doi: 10.1096/fj.06-7294com).

10 Was mitoK channel activity measured in the simulation of these pathologies?

Replay: As replayed to comment 2, we did not measured channel activity.

11 Line 123, 173-175, 200-202. n = 8 biological replicates per group; n ≥ 4 biological replicates per group; n = 3 biological replicates per group; n = 4 biological replicates per group, n = 3 biological replicates per group. What do you mean? Eight measurements per animal? Etc.

Replay: With biological replicates we intend measurements made on different (n) samples. Differently, technical replicates are repeated measurements of the same samples. In any case, to avoid confusion, we have deleted the expressions “biological replicates” in all the figures in which the sample size was indicated as “n=X biological replicates”.

12 Line 221. In the myocardium of IR rats, the early recovery of euthyroidism is accompanied by the maintenance of the post-ischemic levels of mitoK and mitoSur transcripts, which is in line with the improvement in both short-term and medium-term cardiac function previously reported in this preclinical model [13-15]. Can some specific details be given here so that you can evaluate this particular article, rather than looking in references 13-15?

Replay: In accordance with the reviewer’s suggestion some more details has been added in the revised version as follows: “which is in line with the previously reported improvement in both short-term (3d) and medium-term (14d) morpho-functional parameters including LV end systolic diameter, ejection fraction and fractional shortening, and resulted in long-lasting restoring of cardiac performance and geometry [13-15].” Lines 230-234.

13 Line 229. Thus, we propose, that at least in part, the T3 cardiac antioxidant effect is dependent on upregulation of mitoK-ATP. In what way? If possible, make suggestions.

Replay: As suggested by the reviewer we rephrase the discussion section to make our suggestion more explicit as follows: “On the other hand, managing oxidative stress damages in the early stages of IR is a primary goal of cardioprotection to limit the loss of cardiomyocytes and blunt adverse cardiac remodeling. In the post-IR setting, mitochondria are both the primary source and main targets of reactive oxygen species (ROS). In turn, mitochondrial dysfunctions trig-gered by ROS and accumulation of calcium in the mitochondrial matrix, favors cell death and inflammation [17]. Activation of MitoK-ATP under stress conditions has been suggested to cause slight mitochondrial uncoupling, which prevents ROS generation through the reverse electron transport mechanism and limits calcium overload [7; 21-22]. Therefore we propose that T3 replacement favors the antioxidant activity of mitok-ATP by upregulating the subunits of the channel under stress condition”. See lines 240-249.

14 Line 260. The LT3S we observed in or model. A misprint?

Replay: The typo has been corrected as suggested.

Reviewer 2 Report

In this manuscript, Canael et al. used a rat model to study heart damage. The topic of this work is of interest to IJMS, and it uncovers novel molecular mechanisms of T3-dependent cardiac mitoK-ATP expression. The authors first confirm that T3 restores mitoK and mitoSur expression levels upon rat heart damage in this manuscript. To seek putative therapeutic treatment, the authors found that T3 acts as a guard to the full functionality of cardiac cells by maintaining a fine-tuned ratio of mitoSur/mitoK. Overall, I find this work is solid and important to the field. I hope the author can discuss other potential explanations for the T3-sensitive miRNA: mir-133a. It is well-known that miRNAs usually have multiple targets. Would this miRNA also have predicted targets? Will there be any cross-talk between mitoK-ATP and other pathways? I suggest a minor revision, and I look forward to the revised manuscript.

Author Response

Comments and Suggestions for Authors

In this manuscript, Canale et al. used a rat model to study heart damage. The topic of this work is of interest to IJMS, and it uncovers novel molecular mechanisms of T3-dependent cardiac mitoK-ATP expression. The authors first confirm that T3 restores mitoK and mitoSur expression levels upon rat heart damage in this manuscript. To seek putative therapeutic treatment, the authors found that T3 acts as a guard to the full functionality of cardiac cells by maintaining a fine-tuned ratio of mitoSur/mitoK. Overall, I find this work is solid and important to the field.

Replay: We have really appreciated the positive remarks of the reviewer and are grateful for his/her constructive comments and suggestions.

1 I hope the author can discuss other potential explanations for the T3-sensitive miRNA: mir-133a. It is well-known that miRNAs usually have multiple targets. Would this miRNA also have predicted targets? Will there be any cross-talk between mitoK-ATP and other pathways? I suggest a minor revision, and I look forward to the revised manuscript.

Replay: We agree with the reviewer that mir-133a, as all miRNAs, may have multiple targets. Some of the predicted mir-133a targets involved in mitochondrial function and tissue remodeling have already been described in reference 13. The question posed by the reviewer on the cross talk between mitoK-ATP and other pathways is the same that we asked ourselves and which will be the subject of our future investigations. However, at the moment we don't feel confident enough to anticipate any preliminary results. Anyway, the relevance of the reviewer’s issue has been underlined in the revised version, lines 330-331, as follows: “However, it cannot be excluded that other pathways, evoked by the T3-sensitive mir-133a, might influence the expression of mitoK and mitoSur”.

Round 2

Reviewer 1 Report

Thanks to the authors for the clear and detailed answers to the comments